# Synthesis and Characterization of *Konjac Glucomannan*/*Carrageenan*/Nano-silica Films for the Preservation of Postharvest White Mushrooms

**DOI:** 10.3390/polym11010006

**Published:** 2018-12-21

**Authors:** Rongfei Zhang, Xiangyou Wang, Juan Wang, Meng Cheng

**Affiliations:** Department of Food Science and Engineering, Shandong University of Technology, Zibo 255000, China; shenggongrongfei@163.com (R.Z.); wangjuan7912@163.com (J.W.); chengmeng0110@163.com (M.C.)

**Keywords:** nano-silica, preservation, films, carrageenan, konjac glucomannan, white mushrooms

## Abstract

In this study, the *konjac glucomannan* (KGM)/*carrageenan* (KC)/nano-silica film was prepared and characterized by scanning electron microscopy (SEM), Fourier-transform infrared spectroscopy (FTIR), and X-ray diffraction (XRD). The preservation quality of white mushrooms (*Agaricus bisporus*) packed using the films was also determined. The nano-silica dosage was found to affect the properties of the nanocomposite KGM/KC films. The results indicated that the properties of the films were significantly improved with the addition of nano-silica. The water vapor permeability, water solubility, moisture absorption, and light transmittance of KGM/KC/nano-silica films were significantly affected by the nano-silica dosage. In this study, the optimal nano-silica dosage to incorporate into the film in order to achieve excellent performance was 0.3%. Strong intermolecular hydrogen bonds were also observed between KGM/KC and nano-silica in the KGM/KC/nano-silica film by FTIR. In addition, the KGM/KC/nano-silica film markedly reduced the browning index, delayed the weight loss and softening, and extended the shelf life of mushrooms during storage at 4 °C. The KGM/KC film modified using nano-silica can provide a potential method for improving the preservation quality of white mushrooms during storage.

## 1. Introduction

Food packaging is mostly produced from plastic materials. However, plastic materials are resilient to degradation, which could lead to a severe environmental problem. The use of biopolymers has drawn increasing attention for their application in the prevention of serious environmental problems. Biopolymers, as an important renewable energy resource, can provide raw materials for future industrial development [1,2]. In food packaging, films prepared using biopolymers can control the migration of humidity, O_2_, CO_2_, and aromas, and can also carry food ingredients to maintain food quality [3].

*Konjac glucomannan* (KGM) exhibits excellent biocompatibility, film-forming ability, and biodegradability [4,5], and for this has attracted significant interest as a potential packaging material amongst natural biopolymers. *Carrageenans* (KC) have 3,6-anhydro-d-galactopyranose residues and a negatively charged sulfate group in the chain. The characteristics of carrageenan provide its gel-forming ability [6]. Blending KGM/KC films is a convenient and effective method which can improve their physicochemical properties. However, KGM/KC films also have many disadvantages, including inferior mechanical properties, low strength, low water resistance, and poor antimicrobial activity; thus, its food preservation efficiency is unsatisfactory.

Nanotechnology is widely applied in the food industry, including in food processing, food packaging, and food evaluation [7,8,9]. As opposed to substances of normal size, nano-sized substances show quanta-size, small-size, surface, and macroscopic quanta-size effects [10]. Nanoparticles can provide the possibility of enhancing mechanical properties and permeability by taking advantage of their peculiar ability to disperse uniformly in the KC/KGM films and generate a hydrogen bond with KGM/KC molecules through surface hydroxyl groups [11]. Several nano-oxide particles, such as nano-silica, nano-TiO_2_, and nano-ZnO, have been developed as potential materials for nanocomposites [12,13,14]. Nano-silica cannot be digested by the digestive tract and is thus associated with food safety concerns. In addition, nano-silica can be regarded as a food additive [15]. Mixing nano-silica into KGM/KC films may considerably extend the shelf life of food by protecting them from environmental factors (oxygen, light, and bacterial spoilage) [16], restrict enzyme immobilization [17], and inhibit fruit and vegetable decay [18]. Therefore, KGM/KC/nano-silica composite films, a type of biomaterial, can be applied in fruit storage. We found that nano-silica could enhance the preservation properties of KGM/KC coatings to prolong the shelf life of white mushrooms by using coating treatment [19]. However, whether nano-silica dosages could affect the preservation properties of KGM/KC films, with their property, permeability, and microstructure all considered, has yet to be determined.

In this study, a KC/KGM/nano-silica film was prepared and then characterized by SEM, FTIR, and XRD. The preservation properties of KGM/KC/nano-silica films were investigated by packaging the white mushrooms. To obtain excellent properties, the optimal nano-silica dosage was determined. The KGM/KC film modified using nano-silica can provide a potential method for improving the preservation quality of white mushrooms during their storage.

## 2. Materials and Methods

### 2.1. Materials

Nano-silica (20 nm), KC (κ-carrageenan, density = 1.000), and KGM (viscosity ≥30,000; expansion coefficient >100) were purchased from Shanghai Aladdin Biochemical Technology Co., Ltd., Shanghai, China. White mushrooms were purchased in ZiBo City in Shandong Province, China. The mushrooms were then pretreated in preparation for package testing. The mushrooms (*Agaricus bisporus*) had no mechanical damage, blemishes, or disease. Subsequently, they were immediately transported to the Shandong University of Technology to precool for 24 h at 2 °C.

### 2.2. Preparation of Composite Films

KGM/KC films and KGM/KC/nano-silica films were synthesized using the method illustrated in Figure 1. The contents (%) of the materials were evaluated relative to the volume of the solvent (*w*/*v*). For the preparation of the films, nano-silica at varying amounts (0% *w*/*v*, 0.1% *w*/*v*, 0.2% *w*/*v*, 0.3% *w*/*v*, 0.4% *w*/*v*, 0.5% *w*/*v*), 0.6% *w*/*v* KC, 0.2% *w*/*v* KGM, and 0.7% *w*/*v* glycerol were dissolved in the distilled water at 60 °C for 20 min. Subsequently, the mixture was sonicated using an ultrasonic processor (VCX 750, Sonics & Materials, Inc., Newtown, CT, USA) under the following conditions: frequency, 20 kHz; power, 150 W; and duration, 20 min. The 80 mL solution was poured into each glass, which was of equal size (25 cm × 25 cm). Films on the surface of each glass were dried in a circulation oven at 50 °C [(relative humidity (RH) of 50%)]. Ultimately, the films were uncovered for subsequent characterization and measurement of their properties. Prior to characterization of their properties, all films were conditioned at 23 °C and 50% RH for 24 h in a chamber (BSC-150, Shanghai Boxun Co., Ltd., Shanghai, China).

### 2.3. Characterization of Films

Fourier-transform infrared spectroscopy (FTIR): The functional groups were estimated using a Nicolet 5700 analyzer, Thermo Electron Scientific Instruments Corp., Waltham, MA, USA. The spectra of the films were obtained within a 4000–400 cm^−1^ wavelength range [20].

X-ray diffraction (XRD): The crystalline properties of the films were characterized using a D8 Advance analyzer (Germany Bruker Corp., Heidelberg, Germany) equipped with Cu *K*α radiation [21]. The scattering angle was in the 5–45° range with a scan rate of 1°/min.

Scanning electron microscopy (SEM): The morphology of the surface of the films was observed by scanning electron microscopy (SEM, FEI Sirion 200, FEI, Hillsboro, OR, USA) operated at 20 kV. The samples were sputtered with a layer of gold.

### 2.4. Properties of Films

#### 2.4.1. Thickness

The thickness of the films was measured using a micrometer (Dial thickness gauge 7301, Mitutoyo Co., Kawasaki, Japan) with a precision of 0.01 mm, and each film was chosen in at least ten random locations [22].

#### 2.4.2. Water Vapor Transition Rate (WVTR)

The water vapor transition rate (WVTR) of the films was estimated in accordance with the Chinese National Standard GB1037-88 (1998). Dehydrated calcium chloride amounting to 0.3 g was placed in a cup. The films were fixed on the mouth of the cup and sealed with a rubber band. They were then weighed every 1 h for 24 h at 25 °C and RH 90%. The WVTR was evaluated using the following formula:
$$\mathrm{WVTR}=\frac{m_{f}-m_{i}}{D\times S}$$
where *m*_f_ is the weight of the final cup; *m*_i_ is the weight of the initial cup; *D* is the duration, in days; and *S* is the effective area of the films (the area of the cup mouth), in m^2^.

#### 2.4.3. Tensile Strength (*T*_s_)

The tensile strength (*T*_s_) of the films was measured using a texture analyzer (TMS-2000, FTC, Los Angeles, CA, USA). The films were cut into 50 mm × 15 mm rectangular strips. The probe of the Acoustic Multiple Tensile Grips (AMTG) was employed at an initial distance of separation with 30 mm and a velocity of 10 mm/s. All tests were repeated three times.

#### 2.4.4. Color

The color of the film samples was determined using the Chroma Meter (Konica Minolta, CR-400, Tokyo, Japan) with the D65 standard illuminant. All samples (3 cm × 5 cm) had the white standard color plate (L* = 94.6, a = −0.62, and b = 1.42) as the background [23]. The total color difference (Δ*E*) was determined using the following equation:
$$∆E=\sqrt{{\left(L^{*}-L\right)}^{2}+{\left(a^{*}-a\right)}^{2}+{\left(b^{*}-b\right)}^{2}}$$
where L, a, and b denote the differences in each color value of the films.

#### 2.4.5. Transparency

A UV spectrophotometer (UV-2550, Shanghai, China) was used to evaluate the transparency of the films [24]. Film samples were cut into a uniform size (1 cm × 4 cm) and then scanned at a wavelength of 600 nm (*T*_600_).

#### 2.4.6. Water Solubility (WS) and Moisture Absorption (MA)

The procedures for determining WS and MA have been described in detail in our previous study [19]. Samples (3 cm × 5 cm) were cut from each type of film to determine the WS of the films. The films (*m*_1_) were pretreated at 50 °C for 24 h and then immersed in distilled water (30 mL), with occasional stirring at 25 °C for 6 h. Undissolved films (*m*_2_) were dried at 50 °C to a constant weight [11]. The analytical method of determining MA was similar to that described by Moreno et al. (2017) [25]. The films (*m*_3_) were pretreated at 50 °C for 24 h and then placed at 23 °C and RH 60% for 6 h. The films were then weighed (*m*_4_). WS and MA were calculated using the following formulas:
$$\mathrm{WS}=\frac{m_{1}-m_{2}}{m_{1}}\times 100$$
$$\mathrm{MA}=\frac{m_{4}-m_{3}}{m_{3}}\times 100$$

#### 2.4.7. Oxygen Transmission Rate (OTR) and Carbon Dioxide Transmission Rate (CDTR)

The oxygen transmission rate (OTR) and carbon dioxide transmission rate (CDTR) were determined using the method described by Wang et al. (2014) [21]. Different films were fixed on a cup mouth containing 3 g deoxidizer/5g KOH and then sealed with a rubber band. They were then placed under the following conditions: temperature, 23 °C; RH, 90%; and duration, 48 h. Subsequently, the cups were weighed for determination using the equations described in detail by Zhang et al. (2018) [19]. The OTR and CDTR were calculated according to the following equations:
$$\mathrm{OTR}=\frac{∆m_{1}}{D\times S}$$
$$\mathrm{CDTR}=\frac{∆m_{2}}{D\times S}$$
where ∆*m*_1_ is the amount of O_2_ absorbed by the deoxidizer; Δ*m*_2_ is the amount of CO_2_ absorbed by KOH; *D* is the storage time, in days; and *S* is the effective area of the films, in m^2^.

### 2.5. Application of Films for Mushroom Storage Stability

White mushrooms were placed on 1050 mL polypropylene trays and then covered with a KGM/KC/nano-silica film and a KGM/KC film. Each tray contained 160 g samples. Mushrooms without packaging were used as the control. The weight loss, firmness, whiteness, membrane permeability, and sensory evaluation (hardness, fracturability, adhesiveness, stringiness) of the mushrooms were measured at intervals (0, 3, 6, 9, and 12 d) during storage at 4 °C.

Weight loss was measured gravimetrically during storage. The whiteness was measured with a colorimeter (SC-80C, Beijing, China). The tissue electrolyte leakage was determined to obtain the membrane permeability [26]. The firmness was evaluated using a penetrometer (GY-1, Beijing, China). The sensory properties (hardness, fracturability, adhesiveness, stringiness) of the mushrooms were measured with a texture analyzer (TMS-2000, Los Angeles, CA, USA) under the texture profile analysis (TPA) pattern [27,28]. The mushrooms were cut into cuboid shapes (measuring 30 mm × 30 mm × 30 mm) and compressed using a P/36R probe at a speed of 0.5 mm/s and at 30% of compression. From the TPA curves, the following texture parameters were obtained: Hardness at 30% of deformation, fracturability, adhesiveness, and stringiness. Each test was repeated at least three times to ensure reproducibility.

### 2.6. Statistical Design

All experiments were conducted at least three times for each sample. SEM, XRD, and FTIR were conducted without repeat tests. The Statistical Product and Service Solutions (SPSS) software was used to analyze the variance of all data. Duncan’s test indicated significant differences between means at *p* < 0.05.

## 3. Results and Discussion

### 3.1. FTIR Analysis

FTIR spectroscopy was used to evaluate the miscibility of polymers by changes in chemical bonds and groups between materials. The FTIR of films at 4000–400 cm^−1^ are presented in Figure 2. The main chain elements of the KGM/KC/nano-silica films and the KGM/KC films were similar to one another. The wide peaks at 3330 cm^−1^ correspond to the O–H stretching vibration, and the peaks at 2905 cm^−1^ can be attributed to the C–H stretching vibration of the clean KGM/KC film. We observed a similar occurrence in several other polysaccharides, such as chitosan and curdlan [29,30].

Major bands showed significant changes at 3400–3200 cm^−1^ with respect to the positions of different bands of films. For the KGM/KC film and the KGM/KC/nano-silica film, the strong absorption band was transferred to a low wave number. Specifically, the absorption peak of –OH of the KGM/KC/nano-silica film (S3) shifted to 3348 cm^−1^. This shift could be attributed to the hydrogen bonds that formed when the free –OH of nano-silica combined with that of KC/KGM, which was similar to the reports by Yao et al. (2011) [31]. In addition, the peaks around 1000 cm^−1^ was attributed to the bending vibration and stretching vibration of Si–O, Si–H, and Si–C, indicating that hydrogen bonds were formed between the nano-silica and KGM/KC in the films. The FTIR results indicated that the miscibility between KGM/KC and nano-silica was attributed to intermolecular hydrogen bonds and the synergistic interaction of polymers during film formation and blending [12].

### 3.2. XRD Analysis

XRD analysis can show the crystalline nature of films to characterize their compatibility. The XRD results of the KGM/KC films and the KGM/KC/nano-silica films are presented in Figure 3. A broad peak appeared near 22° for films with a small peak. A weak peak appeared at 30° which was attributed to inorganic salts, such as KCl in the KGM/KC films. The XRD pattern changed with the addition of nano-silica. The intensity of the sharp peak at 30° gradually decreased, and only the broad amorphous band was observed. The peak at 30° disappeared when 0.3% nano-silica was incorporated into the films once the salts involved in the interaction formed between KC and KGM. The diffraction peak of the KGM/KC/nano-silica films (S3) became lower and broader than the other films. The results indicated that 0.3% nano-silica could be easily distributed into the KGM/KC matrix and destroy the primal crystalline domains of KGM/KC and nano-silica [23]. XRD showed that nano-silica could possibly interact with KGM and KC because of the –OH or –COO– groups in KGM/KC and –OH on the surface of the nano-silica. Therefore, KC and KGM exhibited excellent compatibility with nano-silica in the films.

### 3.3. SEM Analysis

Many properties of materials in the films could be presented by the morphology of the top surface of the films. The morphological properties of the KGM/KC films and the KGM/KC/nano-silica films were determined by SEM, as shown in Figure 4. All KGM/KC/nano-silica films exhibited graininess, similar to the results obtained by Tang [32] and Wu [33]. With the increase in nano-silica, the size and amount of the particles were increased, and severe particle aggregation occurred at a nano-silica dosage of 0.5%. The particle aggregation was found to be able to limit the improvement of the properties of packing. This finding was consistent with the XRD and FTIR results, which indicated that KGM/KC/nano-silica films (S3) exhibited good dispersibility.

### 3.4. Color and Transparency Analysis

The colors of packaging materials are significant in fruits and vegetables because they directly influence consumer acceptability [34]. The total color differences (∆*E*) and transparency of the KGM/KC and KGM/KC/nano-silica films are listed in Table 1. The ∆*E* of films decreased, and then increased significantly (*p* < 0.05) with the nano-silica dosage. The color of the films exhibited the tendency to change into transparency. The color of films acceptable to consumers was obtained at a nano-silica content of 0.1%–0.3%. Ultraviolet light easily affects the quality of food products in food packaging. The opacity of films is also an important factor. High opacity indicates less transparency [35]. The transparency of the composite films decreased with the nano-silica content. The KGM/KC/nano-silica (S3) films exhibited the highest opacity, indicating that it could potentially enhance the UV-barrier property owing to the special optical effect of nano-silica [36]. The nano-silica with a wavelength smaller than that of light exerted certain optical effects. Therefore, ultraviolet light could not easily pass through the low-transparency film. In addition, the nanocomposite films exhibited low transparency with increasing nano-silica [30,37].

### 3.5. Tensile Strength Analysis

The *T*_s_ of the KGM/KC/nano-silica films increased with the addition of nano-silica, and then decreased significantly (*p* < 0.05). The optimal mechanical properties (69.11 MPa) were obtained at 0.3% nano-silica (Table 1). Excessive amounts of nano-silica might have led to a sharp decrease in the dispersion of nano-silica. The results were consistent with SEM.

### 3.6. Film Thickness Analysis

The thicknesses of the films showed no significant difference (*p* > 0.05) with the addition of nano-silica. Thickness uniformity can help examine the comprehensive performance of films.

### 3.7. WVTR Analysis

The WVTRs of the KGM/KC and KGM/KC/nano-silica films are summarized in Table 2. The KGM/KC/nano-silica films exhibited lower water vapor permeability (WVP) results than those of the KGM/KC films. Plasticizers could enhance the permeability of films, such as water vapor, gas, and solute permeability [38]. However, the number of plasticizers (glycerol) remained the same. Thus, the difference was attributed to the addition of nano-silica in the films. The results showed that the WVTR decreased with the addition of nano-silica and then increased significantly (*p* < 0.05). The optimal WVP (644.41 g·m^−2^·d^−1^) was obtained at 0.3% nano-silica. The dispersed nanoparticles changed the micropore structure of the KGM/KC films; thus, the WVTR decreased with the increase in nano-silica. However, when nano-silica content became excessive, some of the particles were severely agglomerated owing to non-uniform dispersion [38]. Nano-silica enhanced the physical crosslinking between KC and KGM, which changed the seepage path of water molecules in the films and suppressed the penetration of water.

### 3.8. WS and MA Analysis

The WS and MA of the KGM/KC and KGM/KC/nano-silica films are listed in Table 2. WS can be used to measure the films’ water resistance. The WS of the KGM/KC/nano-silica films decreased with an increase in nano-silica content. These results indicated that the KGM/KC/nano-silica films (S3) exhibited the highest water resistance. The hydrophilicity of the polymer could influence the WS of the films. MA was used to evaluate the moisture retention of the films and presented in reverse order. The MA of the KGM/KC/nano-silica films increased with an increase in nano-silica content, which could be attributed to the large amounts of –OH on the surface of the nano-silica and the hydrogen bonding between the nano-silica and water, thereby increasing the water-holding ratio of the nanocomposite films. Specifically, the MA of the KGM/KC/nano-silica films (S3) markedly increased to 297%. The reason for this could be that 0.3% nano-silica was better dispersed in the films than 0.4% nano-silica and 0.5% nano-silica to form the tight structure, which was supported by the SEM. Thus, an increase in film water resistance could be suitable for food packaging under highly humid conditions.

### 3.9. OTR and CDTR Analysis

Oxygen can cause many degradation reactions in foods, which shortens their shelf-life. Low oxygen permeability of food packaging materials was found to play a crucial role in preservation [39]. Table 2 shows that the gas transmission rate (OTR and CDTR) decreased with the addition of nano-silica. Among the films, the KGM/KC/nano-silica films (S3) exhibited the lowest OTR and CDTR. During packaging, a violent movement in the macromolecular chains of the films was observed, which was caused by the difference in pressure. Gas (O_2_ and CO_2_) passed through the channel and were released from the side with low partial pressure [12] Nano-silica at 0.3% was dispersed evenly, which changed the micropore structure of the KGM/KC films, allowing the adjustment of O_2_–CO_2_ exchange by the Si-O-Si groups of nano-silica in the films.

### 3.10. Application of Films for Mushroom Storage Stability

The storage stability of the films was evaluated by using them to pack fresh white mushrooms. These mushrooms were packaged using the KGM/KC and KGM/KC/nano-silica films (S3). The mushrooms with packaging without any film at 4 ± 1 °C was considered as the control group. After a storage period of 12 d, the appearance of the treated mushrooms was evaluated (Figure 5). The mushrooms in the control packaging appeared to have severely wilted. *Agaricus bisporus* packed using the KGM/KC films had significantly wilted, whereas those packed using the KGM/KC/nano-silica films remained fresh on the surface.

Changes in the weight loss, whiteness, firmness, and membrane permeability of the mushroom samples are illustrated in Figure 6. The weight loss and membrane permeability of the control samples were the greatest, whereas those of the mushrooms treated with the KGM/KC/nano-silica films (S3) were the least. The reason for this was mainly that the WVP of the KGM/KC/nano-silica films (S3) was lower than that of the KGM/KC films. Moreover, the whiteness and firmness of the samples treated with the KGM/KC/nano-silica films (S3) were of the highest degree (Figure 6). A reason for this could be that the silicon–oxygen bond in the KGM/KC/nano-silica films could have affected the absorption, dissolution, diffusion, and release of CO_2_/O_2_. The CO_2_/O_2_ exchange capacity inside and outside the films could be adjusted to inhibit the respiration of mushrooms.

The hardness, fracturability, adhesiveness, and stringiness of the mushrooms were measured to study the change in their texture. For the mushroom samples, the fracturability, hardness, and springiness were reduced because of senescence and metabolism. As shown in Figure 7, the mushrooms treated with the KGM/KC/nano-silica films exhibited significantly higher levels of hardness and fracturability (*p* < 0.05) compared with other treatments. The lowest levels of hardness and fracturability for the control group were obtained on Day 12 of storage. Meanwhile, the adhesiveness and stringiness of the samples packaged with the KGM/KC/nano-silica films decreased (Figure 7). The effect of the KGM/KC/ nano-silica films on the quality preservation of white mushrooms could be attributed to the slow metabolic activity and low respiration, which delayed senescence. The nano-silica polymer films formed a super-semipermeable film around the mushrooms, which changed the internal atmosphere by increasing CO_2_ and reducing O_2_, thus inhibiting ethylene evolution [40].

## 4. Conclusions

In this study, the properties of a blend biopolymer film added with nano-silica at varying proportions were investigated. The nano-silica dosage affected the properties of the nanocomposite KGM/KC films. The water vapor permeability, water solubility, moisture absorption, and light transmittance of KGM/KC/nano-silica films were significantly affected by the nano-silica dosage. The optimal nano-silica dosage that had to be incorporated into the films in order to achieve excellent performances was 0.3%. Strong intermolecular hydrogen bonds were also observed between KGM/KC and nano-silica in the KGM/KC/nano-silica film by FTIR. The KGM/KC/nano-silica films (S3) showed a tendency to extend the shelf life of fresh *Agaricus bisporus* from 5 d to 12 d, maintaining their whiteness, visual appearance, and hardness. The KGM/KC/nano-silica films involved simple processing and exhibited industrial feasibility. Therefore, the KGM/KC/nano-silica films in the packaging of *Agaricus bisporus* was found to be able to control postharvest metabolism and senescence to extend the mushrooms’ shelf-life and maintain their quality. The films can provide a potential alternative to conventional food packaging, and the development of the KGM/KC/nano-silica films can also broaden the application of nanotechnology in the food industry.

## Figures and Tables

**Figure 1 polymers-11-00006-f001:**
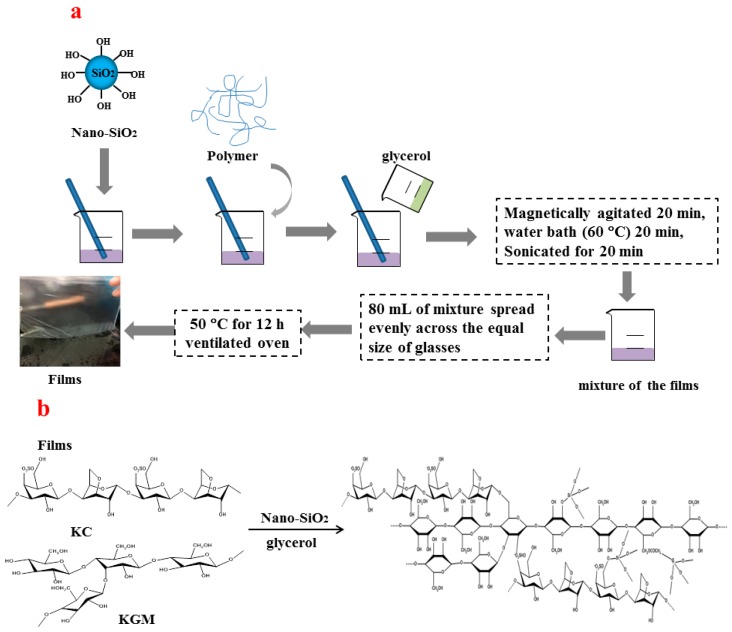
(**a**) Schematic of the films’ preparation, and (**b**) the structure of the obtained films.

**Figure 2 polymers-11-00006-f002:**
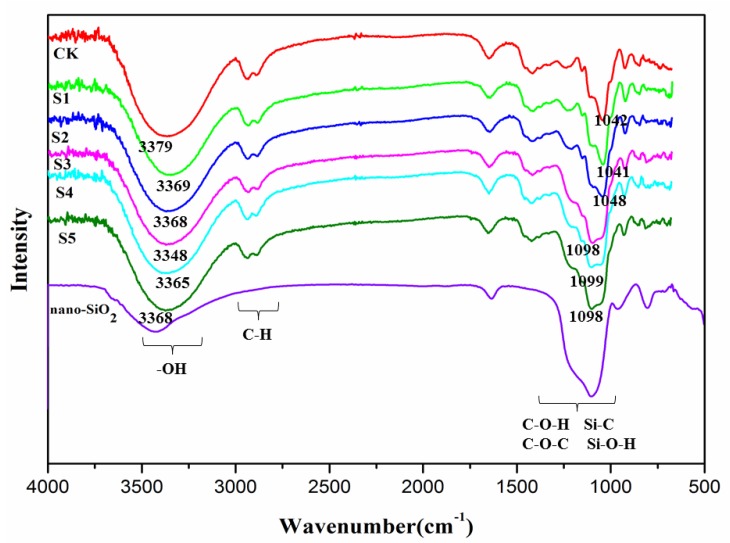
Fourier-transform infrared spectroscopy (FTIR) spectra of blend films (CK: 0% nano-silica; S1: 0.1% nano-silica; S2: 0.2% nano-silica; S3: 0.3% nano-silica; S4: 0.4% nano-silica; S5: 0.5% nano-silica).

**Figure 3 polymers-11-00006-f003:**
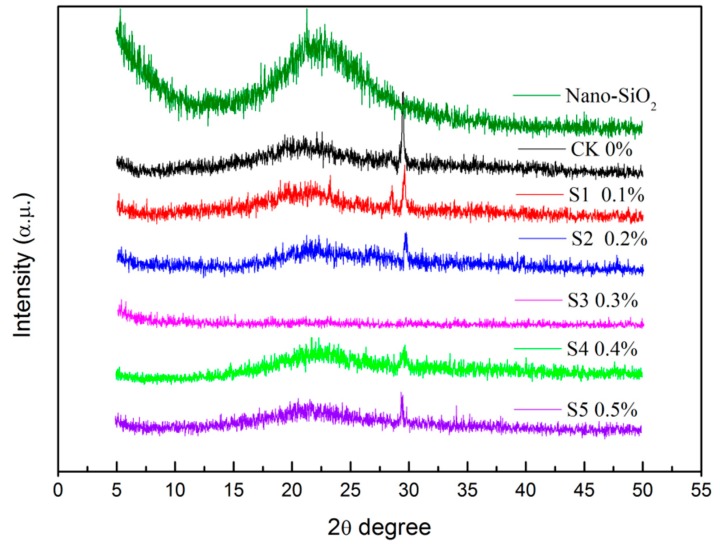
X-ray diffraction (XRD) of *konjac glucomannan* (KGM)/*carrageenan* (KC)/nano-silica films and KGM/KC films (CK: 0% nano-silica; S1: 0.1% nano-silica; S2: 0.2% nano-silica; S3: 0.3% nano-silica; S4: 0.4% nano-silica; S5: 0.5% nano-silica).

**Figure 4 polymers-11-00006-f004:**
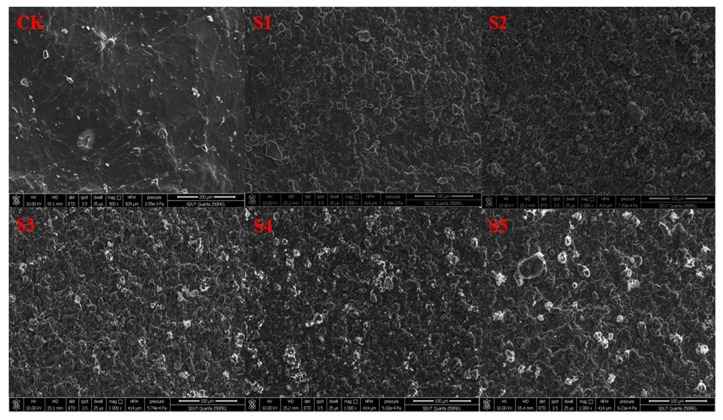
Scanning electron microscopy (SEM) images of KGM/KC films (CK) and KGM/KC/nano-silica films with different nano-silica dosages. (CK: 0% nano-silica; S1: 0.1% nano-silica; S2: 0.2% nano-silica; S3: 0.3% nano-silica; S4:0.4% nano-silica; S5:0.5% nano-silica).

**Figure 5 polymers-11-00006-f005:**
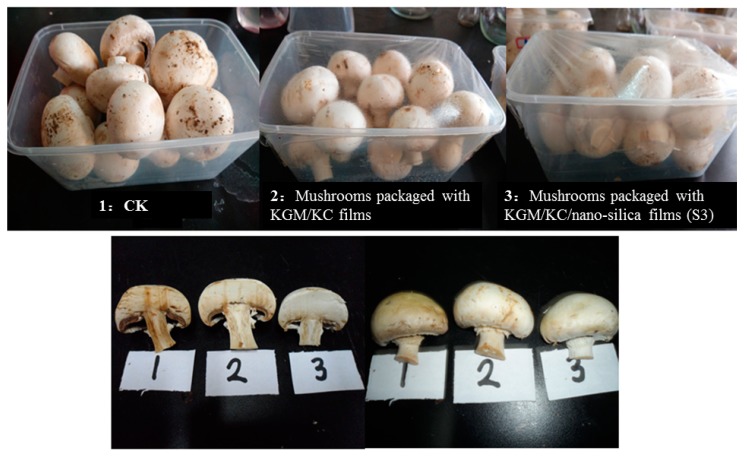
Effect of film packaging with *Agaricus bisporus* stored at 4 ± 1 °C for 12 d (1: CK; 2: mushrooms packaged with KGM/KC films; 3: mushrooms packaged with KGM/KC/nano-silica films (S3)).

**Figure 6 polymers-11-00006-f006:**
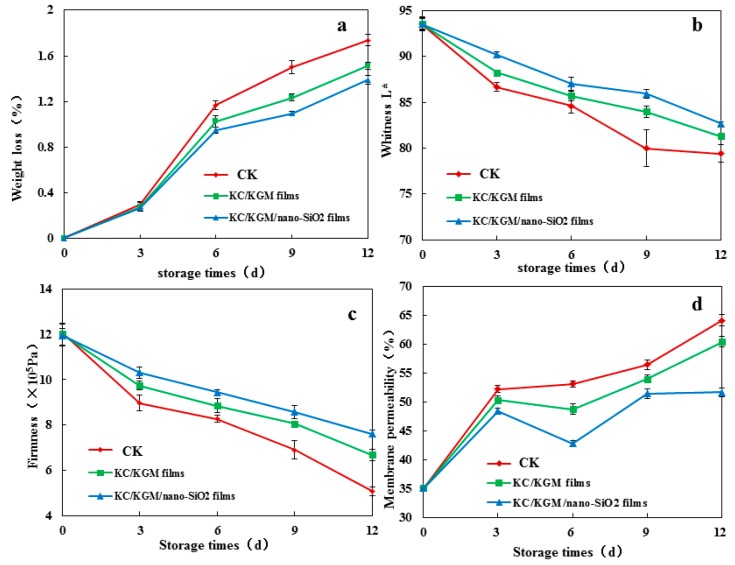
Effect of the films packaged with *Agaricus bisporus* stored at 4 ± 1 °C for 12 d (1: CK; 2: mushrooms packaged with KGM/KC films; 3: mushrooms packaged with KGM/KC/nano-silica films (S3)).

**Figure 7 polymers-11-00006-f007:**
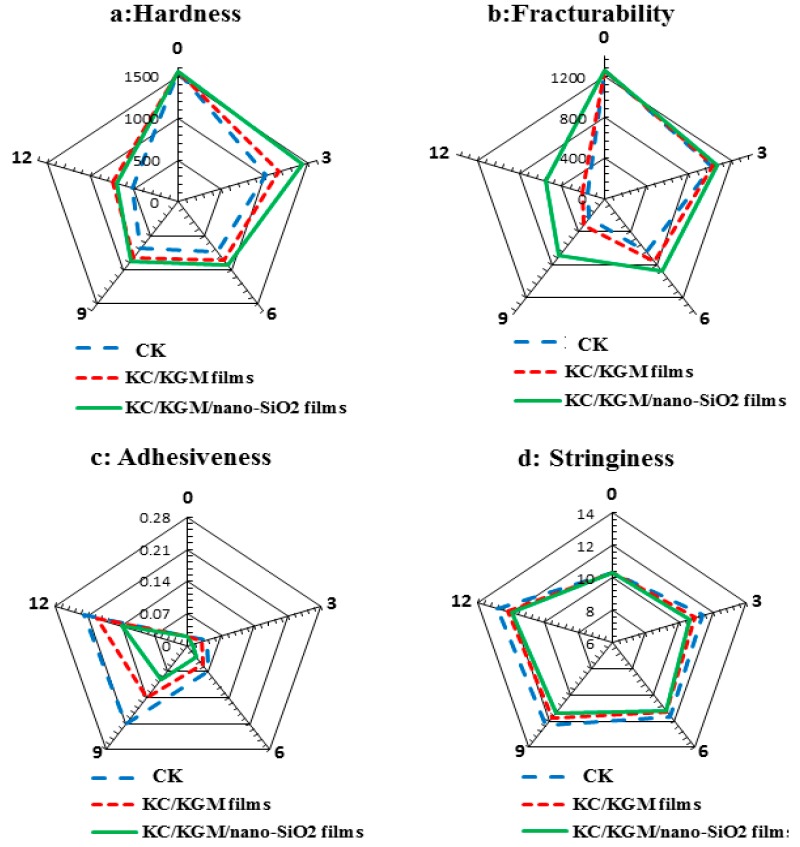
Effects of *Agaricus bisporus* on sensory evaluation by using different treatments during the storage period.

**Table 1 polymers-11-00006-t001:** The total color differences ∆*E*, transparency, and *T*_s_ of KGM/KC films and KGM/KC/nano-silica films.

Films	∆*E*	*T*_600nm_ (%)	*T*_s_ (MPa)
CK	4.02 ± 0.05 ^a^	90.36 ± 0.35 ^a^	50.63 ± 2.1 ^f^
S1	3.81 ± 0.21 ^ab^	83.21 ± 0.24 ^b^	61.32 ± 1.3 ^e^
S2	3.32 ± 0.02 ^c^	80.13 ± 0.15 ^c^	63.51 ± 3.2 ^d^
S3	2.67 ± 0.22 ^f^	73.91 ± 0.17 ^f^	69.11 ± 1.5 ^a^
S4	2.98 ± 0.01 ^e^	74.45 ± 0.21 ^e^	67.22 ± 1.1 ^c^
S5	3.19 ± 0.13 ^d^	75.32 ± 0.09 ^d^	68.01 ± 2.7 ^b^

Different superscripts indicate a significant difference (*p* < 0.05) within the columns.

**Table 2 polymers-11-00006-t002:** Thickness, water vapor permeability (WVP), water solubility (WS), moisture absorption (MA), oxygen transmission rate (OTR), and carbon dioxide transmission rate (CDTR) of films.

Films	Thickness (μm)	WVP (g·m^−2^·d^−1^)	WS (%)	MA (%)	OTR (g·m^−2^·d^−1^	CDTR (g·m^−2^·d^−1^)
CK	26.4 ± 0.48 ^a^	806.4 ± 9.12 ^a^	80.1 ± 0.71 ^a^	176.4 ± 5.88 ^f^	0.039 ± 0.002 ^f^	0.238 ± 0.011 ^f^
S1	25.9 ± 0.31 ^a^	696.3 ± 5.22 ^d^	68.7 ± 0.85 ^b^	213.1 ± 6.17 ^e^	0.030 ± 0.001 ^e^	0.204 ± 0.016 ^cd^
S2	26.3 ± 0.26 ^a^	662.0 ± 3.78 ^e^	61.3 ± 0.32 ^c^	271.2 ± 4.12 ^d^	0.023 ± 0.003 ^b^	0.198 ± 0.008 ^bc^
S3	25.6 ± 0.33 ^a^	644.4 ± 1.44 ^f^	52.4 ± 0.41 ^f^	297.3 ± 3.25 ^a^	0.016 ± 0.002 ^a^	0.153 ± 0.021 ^a^
S4	26.9 ± 0.21 ^a^	743.6 ± 2.32 ^c^	59.2 ± 0.56 ^cd^	280.5 ± 1.36 ^c^	0.026 ± 0.001 ^c^	0.187 ± 0.009 ^b^
S5	26.2 ± 0.39 ^a^	783.8 ± 4.26 ^b^	58.4 ± 0.38 ^de^	286.1 ± 2.47 ^b^	0.028 ± 0.002 ^cd^	0.206 ± 0.006 ^de^

Values are means ± standard deviation. The different letters in a column indicate significant differences at *p* < 0.05.

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
