# Peer review of "Synthesis and Characterization of Konjac Glucomannan/Carrageenan/Nano-silica Films for the Preservation of Postharvest White Mushrooms"

_polymers, 2018, doi:10.3390/polym11010006_

Round 1

Reviewer 1 Report

After the next evaluating of the manuscript Polymers-413063 entitled " Synthesis and characterization of konjac glucomannan /carrageenan/nano-silica films for the preservation of postharvest white mushrooms" by Rongfei Zhang, Xiangyou Wang, Juan Wang and Meng Cheng, I can state that Authors made necessary corrections and supplementations taking under consideration my recommendations. So, I can recommend this paper for publication in Polymers Journal. 

Author Response

Thank you very much for your in-depth reviewing reports and instructions on our paper. On behalf of my co-authors, we would like to express our great appreciation to you.

Thank you and best regards.

Yours sincerely,

Rongfei Zhang, Xiangyou Wang, Juan Wang

Reviewer 2 Report

Dear Authors,

Many thanks for your kind response. The manuscript is fine by me, except that you appear to have missed addressing my request on Figure 5. It is still unclear which image is of the control (i.e. KGM/KC) and which of KGM/KC/nano-silica. Please clarify this in the figure legend.

Author Response

Thank you very much for your in-depth reviewing reports and instructions on our paper. We have added the legend in the Figure 5 on Page 10 (L286) of the paper as follows:

Figure 5. Effect of films packaging with Agaricus bisporus stored at 4 °C±1 °C for 12 d (1: CK; 2: mushrooms packaged with KGM/KC films; 3: mushrooms packaged with KGM/KC/nano-silica films (S3))

This manuscript is a resubmission of an earlier submission. The following is a list of the peer review reports and author responses from that submission.

Round 1

Reviewer 1 Report

Review on manuscript: polymers-400459:

Synthesis and characterization of konjac glucomannan / carrageenan /nano-silica films on the preservation of postharvest white mushrooms.

by Rongfei Zhang, Xiangyou Wang, Juan Wang and Meng Cheng

submitted to Polymers

In the manuscript submitted for comments the Authors prepared and characterized konjac / carrageenan / nano-silica films and studied its potential use as a packing material for white mushrooms.

In my opinion the manuscript fits in the aim and scope of the journal and after major revision could be consider for publication in Polymers.

Detailed recommendation:

lines 12-13 – Italic style should be used,

line 17 – in what respect is optimal?

line 68 – Latin name of mushroom should be added,

lines 67-71 – information about konjac glucomannan and carrageenan should be added,

Figure 1 – the supposed structure of the obtained film should be shown in a separate figure,

line 106 – does d mean days?

line 111 – spaces between words should be used,

line 114 – type of illuminant and measuring geometry should be specified,

line 118 – total color differences should be calculated in relation to the control film and not to the reference plate,

lines 142 and 143 – hardness and hardness? what is the difference?

lines 143 and 148-149 – please explain if the texture parameters were analyzed by sensory analysis method or instrumentally with a texturometer, the parameters analyzed should be defined and the method should be described in detail,  

lines 146 and 148 – description of the methods used or the relevant citations should be added,

line 207 – ∆E means total color differences,

Table 1 – total color differences should be calculated in relation to the control film (CK sample),

lines 210-211 – please explain,

line 226 – if shouldn’t be: columns?

lines 292 – Italic style should be used,

line 308 – see comments above, sensory evaluation of the parameters characterizing the texture is performed with the participation of the evaluation panel and is not the same as the instrumental analysis of the texture.

Reviewer 2 Report

This is a very interesting and well written paper, and my recommendation is to accept it for publication. However, the language needs to improved, and the following minor issues need to be addressed:

Fig 2 (left part) - The identification of the C-H and O-H peaks appear to have been mistakenly switched.

Table 1: Please provide description of the reference substance against which the delta E values were calculated.

Section 3.8: Please add a discussion, if possible, on probable reasons for the higher moisture absorption of the S3 films, considering it had lower silica contents compared to S4 and S5, but presumably similar contents of the two biopolymers.

Figure 5 (bottom part) - In the two sets of mushrooms labeled "1, 2, 3", please indicated which was the control and which the experimental set.

Reviewer 3 Report

This paper needs deep reconsideration, because the drawn conclusions are not fully proven (mainly FTIR and XRD spectra). It is necessary to re-examine the results and re-edit the work. English is not acceptable and the manuscript contains many serious mistakes. Below are examples of sentences that are practically incomprehensible. The text should be checked by a native speaker.

Abstract:

l. 13-15  The nano-silica dosage affects the properties of the nanocomposite KGM/KC films. The results indicated that the properties of the films significantly increased with the addition of nano-silica.

The properties cannot increase. The sentences should be changed.

Introduction:

l. 26-27  The plastic materials are not feasible to be degraded, which might cause a severe environmental problem, which are degraded difficultly.

l. 43-46  Nano particles could provide well the possibility of mechanical properties and permeability, which takes  advantage of their peculiar properties to disperse uniformly in the KC/ KGM films and generate  hydrogen bond with KGM/KC molecules through surface hydroxyl

l. 60-61  The latent properties of KGM/KC/ nano-silica films packaging on prolonging the shelf-life of white mushrooms was investigated to determine the storage effects of fruits and vegetables

These sentences should be improved, they are not clear.

The sentence: (l. 57) In addition, the preservation test of the films for mushrooms was performed by package treatment should be moved to the end of introduction

Materials and methods

l. 75  dissolved to the  distilled water  should be: dissolved in the distilled water

l. 80-82  Prior to characterization of their properties, all films were conditioned at 23 °C and a relative humidity (RH) of 50% for 24 h in the constant temperature and humidity chamber (BSC-150, China).

in the constant temperature - it is unnecessary information because the temperature is given (23 °C)

l. 106  D was the time, d;    What means d?

l. 121-124  The UV spectrophotometer (UV-2550, China) was used to evaluate the transparency of the films [23]. Films samples were cut into a uniform size (1 cm x 4 cm), and then scanned at a wavelength of 600 nm (T600). T600 was calculated using the following Eq.: T = -logT600/Thickness.

I understand that the value T was calculated, but there is a mistake in the last sentence. It should be improved. The same mistake is in Table 1, where T600nm(%) is given

l. 131-132  The films were then weighed to be calculated. The sentence is a mental shortcut and is not clear

l. 136-137  Subsequently, the glass cups were weighed to be calculated as the detailed equations described by Zhang et al. (2018).  It is not clear and should be explained more precisely.

Results and discussion

FTIR analysis

FTIR analysis is incorrect. Information given in the text is different than those presented in Fig. 2. Authors wrote that the wide peaks at 3330 cm‑1 correspond to the C-H stretching vibration, and the peaks at 2905 cm‑1 can be attributed to the O-H stretching vibration, whereas the reverse description is showed in the Fig. 2. Moreover, it is very difficult to find the shift of a broad band from 3346 to 3330 cm‑1. I also do not understand on which base the Authors claim that the intermolecular hydrogen bonds between KGM/KC and nano-silica increased. In which way did such bonds increase (their strength or their amounts)? The right side of Fig 2 shows the spectra in the range of 650-900 cm-1 but the spectrum of SiO2 is not showed so the Authors’ conclusion that new hydrogen bonds and Si-O were formed between nano-silica and KGM/KC is not proved. What is the meaning of two rectangles in the Figure 2?

XRD analysis

Similarly as for FTIR, the analysis of XRD diffractograms is not well presented and the conclusions are too far-fetched. All spectra exhibited broad, low intensive peak indicating the presence of amorphous material in all samples. Hence, the spectra are not very informative and negligible changes in their intensities can result from the small differences in the amounts of samples analyzed, which has not been controlled. It is very risky to compare the XRD results with SEM images because SEM differentiates strongly the samples, for instance S2 and S5, whereas their XRD spectra are practically the same. Moreover, it is worth noticing that the S3 sample spectrum, which does not show any X-ray reflections, is less noisy, which may suggest lower amplification of the recorded spectrum and thus can explain the lack of any peaks. In my opinion conclusions drawn on the basis of so little differentiating XRD spectra are an over-interpretation.

 Color and transparency analysis 

There is a discrepancy between the text and the data in Table 1. According to the text: ΔE of films increased, and then decreased significantly (P < 0.05) with the dosage of nano-silica, whereas in Table 1 the permanent increase of ΔE is showed. Moreover, the sentence: Nano-silica had certain optical effects because the nano-silica considered was smaller than the wavelength of light should be changed.

WVTR analysis

The KGM/KC/nano-silica films obtained lower WVP compared with the KGM/KC films. The word obtained should be changed

WS and MA analysis

The WS and MA of the KGM/KC and KGM/KC/nano-silica films are listed in Table 2 and d. A mistake: Table 2 and d

l. 249  This difference  - which difference?

OTR and CDTR analysis

During packaging, the macromolecular chains of the films occurred a violent movement, which was caused by the difference in pressure. The gas (O2 and CO2) would pass through the channel and release from the side with low partial pressure [12] 0.3% of  nano-silica were dispersed evenly, which changed the micropore structure of the KGM/KC films  and that the Si-O-Si groups could adjust the exchange of O2 and CO2. The results were consistent with the analysis of FTIR.

The text in not clear, and it is difficult to find the correlation between results presented above and FTIR data 

l. 285-286  The reason was mainly that the WVP of the KGM/KC/nano-silica films  was higher than that of the KGM/KC films. In Table 2 such relation is not visible

Conclusions should be rewritten after revision of the paper.